# Pt/Au Nanoparticles@Co₃O₄ Cataluminescence Sensor for Rapid Analysis of Methyl Sec-Butyl Ether Impurity in Methyl Tert-Butyl Ether Gasoline Additive

**Zhaoxia Shi, Ling Xia \*, Gongke Li \* and Yufei Hu**

School of Chemistry, Sun Yat-sen University, Guangzhou 510006, China; shizhaox3@mail2.sysu.edu.cn (Z.S.); huyufei@mail.sysu.edu.cn (Y.H.)

\* Correspondence: xialing@mail.sysu.edu.cn (L.X.); cesgkl@mail.sysu.edu.cn (G.L.)

**Abstract:** High purity methyl tert-butyl ether (MTBE) can be used to adjust gasoline octane values. However, an isomer, methyl sec-butyl ether (MSBE), is the main by-product of its industrial production, and this affects the purity of MTBE. Pt/Au NPs@Co₃O₄ composites with a hollow dodecahedron three-dimensional structure were synthesized using ZIF-67 as a template, with Pt and Au nanoparticles (NPs) evenly distributed on the shell of the hollow structure. A CTL sensor was established for the determination of MSBE based on the specificity of Pt/Au NPs@Co₃O₄. The experimental results showed that Pt/Au NPs@Co₃O₄ had a strong specific cataluminescence (CTL) response to MSBE, with no interference from MTBE. The linear range was 0.10–90 mg/L, the limit of detection was 0.031 mg/L ($S/N = 3$), the RSD was 2.5% ($n = 9$), and a complete sample test could be completed in five minutes. The sensor was used to detect MSBE in MTBE of different purity grades, with recoveries ranging from 92.0% to 109.2%, and the analytical results were consistent with those determined by gas chromatography. These results indicate that the established method was accurate and reliable, and could be used for rapid analysis of MTBE gasoline additive.

**Keywords:** Pt/Au NPs@Co₃O₄; cataluminescence sensor; methyl sec-butyl ether; methyl tert-butyl ether; rapid analysis; gasoline additive





## 1. Introduction

Octane value is one of the important quality indicators of gasoline [1]. Methyl tert-butyl ether (MTBE) is an effective additive for gasoline octane value blending [2,3]. There are strict quality requirements for MTBE used in gasoline octane blending. China's National Energy Administration has issued industry standards for MTBE used in gasoline blending, which stipulate that the purity of MTBE shall not be less than 96% (wt%). Methyl sec-butyl ether (MSBE) is an isomer of MTBE, and known to be the main by-product of the industrial production of MTBE. Their high structural similarity brings difficulties to their separation and detection. Gas chromatography is the current method for determining the content of MSBE in MTBE when evaluating its purity. However, the operation is tedious and is not conducive to rapid analysis. Therefore, it makes sense to establish a simple and rapid method for monitoring MSBE in MTBE gasoline additive, and for determining the purity of MTBE. However, to date, there is no fast-sensing technology that could be used for the rapid detection of MSBE.

As a quick analysis method, cataluminescence (CTL) has many benefits, such as high selectivity, fast response, and easy operation [4,5]. The catalyst is the core component of a CTL sensor, and it has an important effect on sensing performance. Therefore, a CTL sensor could be established based on gas-sensing catalyst materials that can selectively respond to MSBE, and might be used to detect MSBE in MTBE gasoline additive. In recent years, metal oxides such as WO₃ [6], ZnO [7], SnO₂ [8], and Co₃O₄ [9] have been a common type of material used in various sensors. Owing to its good catalytic performance and

low cost, $Co_3O_4$ is a potential material for this application, and it has been widely studied in many fields [10–12]. In addition, as a p-type semiconductor material, the gas-sensing capability of $Co_3O_4$ is based on the catalytic properties of its surface, and it has been widely used as a catalyst to improve the selective oxidation of various volatile organic compounds (VOCs) [9,13].

As a porous framework consisting of metal ions and organic ligands, metal–organic frames (MOFs) have various topologies and tunable structures [14]. They can convert to different types of metal oxides after heat treatment, and maintain their original form. Based on this characteristic, a MOF could be used as an ideal self-sacrificial template for the synthesis of metal oxides with regular morphologies, and to improve the gas-sensing properties of materials [15]. In addition, to further enhance the sensing signal and to speed up the response recovery time, it has become popular to add a small amount of noble metals, such as Ag, Au, Pt, etc., to metal oxides [16,17]. Loading noble metal nanoparticles onto $Co_3O_4$ not only prevents the agglomeration of the noble metal particles, increasing the loading capacity of the precious metal, but also improves the catalytic activity of $Co_3O_4$. For example, an $Ag@Co_3O_4$ composite was reported to increase the signal from formaldehyde by several times, and to reduce the working temperature of the sensor to 90 °C [18]. Furthermore, the performance of composite material in catalyzing CO oxidation has also been explored by depositing Au NPs onto different oxides [8,19]. Recently, bimetallic catalytic materials have attracted attention for use in many catalytic reactions, and the performance of monometallic catalysts is often improved by the addition of a second metal. Bimetallic catalysts based on metal oxides have become a research hotspot due to their high catalytic activity and selectivity for various catalytic conversion processes [20–22]. The advantages of dual noble metal catalysts stem from their electronic interactions, with the possibility of complex geometric arrangement, charge transfer, or interfacial stabilization between the atoms of the two noble metals. The development of new dual noble metal-modified $Co_3O_4$ composites is of great significance for improving their catalytic performance and for broadening their field of application.

In this work, we synthesized Pt/Au NPs@$Co_3O_4$ double noble metal-modified nanocomposites via a two-step method using ZIF-67 as a template, and we explored their application as catalysts using a CTL method to detect the MSBE content of MTBE. The reaction products of the CTL reaction were analyzed by gas chromatography–mass spectrometry to explore the possible mechanism of the Pt/Au NPs@$Co_3O_4$-catalyzed selective CTL response of MSBE. Ultimately, a Pt/Au NPs@$Co_3O_4$-based CTL sensor was constructed for rapid analysis of MTBE purity.

## 2. Experiment

### 2.1. Chemicals

Cobalt nitrate, 2-methylimidazole, methanol, ethanol, sodium chloroaurate, chloroplatinic acid hexahydrate ($H_2PtCl_6·6H_2O$), $H_2O_2$ (30%, wt%), acetone, N,N-dimethylformamide (DMF), and 95%, 98%, 99%, and 99.9% (wt%) MTBE were obtained from Aladdin Reagent Co., (Shanghai, China). MSBE (99.9%, wt%) was purchased from J&K Chemicals, (Beijing, China). All compounds were from commercial sources and were not purified before use. Throughout the studies, ultrapure water (18.3 MΩ·cm) was supplied from a Milli-Q gradient system (Millipore, Billerica, MA, USA).

### 2.2. Instruments

A Hitachi H-7500 transmission electron microscope (Tokyo, Japan) was used to characterize the morphology and size of Pt/Au NPs@$Co_3O_4$. Cu Kα incident radiation (Rigaku, Japan) was used for X-RAY Diffraction (XRD) testing. An Escalab 250 spectrometer (Thermo Fisher Scientific, Waltham, MA, USA) was used to record the X-ray photoelectron spectroscopy (XPS) spectrum. An ASAP-2020 M gas adsorption instrument (Micromeritics, Atlanta, GA, USA) was used to measure surface area. All the CTL experiments were carried

out using an ultraweak luminescent analyzer with a photomultiplier tube (PMT) acquired from Academia Sinica's Institute of Biophysics (Beijing, China).

### 2.3. Synthesis of Pt/Au NPs@Co$_3$O$_4$

The Pt/Au NPs@Co$_3$O$_4$ composite was prepared according to published methods with minor changes; this mainly included three steps:

Synthesis of ZIF-67 precursor [10]: 1.164 g Co(NO$_3$)$_2$·6H$_2$O and 1.312 g 2-methylimidazole were dissolved in 100 mL methanol, then the solution was rapidly stirred and aged at room temperature for 24 h. The precipitate obtained was washed with ethanol 4 times, and the ZIF-67 precursor was obtained after vacuum drying overnight.

Preparation of Au NPs@Co$_3$O$_4$ [23]: ZIF-67 (50.0 mg) was added to 2 mL ethanol containing 0.12 mmol NaAuCl$_4$·2H$_2$O, and allowed to stand at room temperature (approximately 27 °C) for 20 min. Then, the product was collected by centrifugation, washed with ethanol 3 times, and dried under vacuum. The purple ZIF-67/Au$^{3+}$ precursor was heated to 300 °C at a heating rate of 1 °C/min for 2 h. After calcination, the purple precursor turned into a black powder.

Synthesis of Pt/Au NPs@Co$_3$O$_4$ [24]: The modification of Pt NPs was carried out according to a reported method. 2 g Au NPs@Co$_3$O$_4$ was ultrasonically dispersed in 100 mL aqueous solution containing 0.051 mmol H$_2$PtCl$_6$. 0.5 g 30% H$_2$O$_2$ (wt%) was diluted with 20 mL water, then the solution was added to the Au NPs@Co$_3$O$_4$ suspension dropwise with vigorous stirring, while maintaining the temperature at 60°C. The resulting solid was filtered, washed with water, and dried at 110 °C overnight. Finally, the powder obtained by sintering at 350°C for 4 h in a pure H$_2$ atmosphere was denoted as Pt/Au NPs@Co$_3$O$_4$.

### 2.4. Cataluminescence Method

A pulp suspension was obtained by mixing 0.8 g Pt/Au NPs@Co$_3$O$_4$ with 3.0 mL deionized water. After that, a certain volume of pulp suspension was smeared onto the surface of a ceramic heating rod and annealed in air to generate a catalyst layer. The ceramic heating rod coated with the catalyst layer was inserted into a home-made quartz tube with an air inlet and outlet, to form a CTL reaction chamber. The ceramic heating rod was connected to a voltage regulator that was used to control the working temperature by adjusting the output voltage. A BPCL ultra-weak luminescence analyzer was used to detect and record the CTL signal. MTBE gases of different concentrations were obtained by injecting different volumes of MSBE standard solutions into 1 L Teflon sample bags. Unless otherwise specified, the detection conditions were as follows: temperature: 187 °C, wave length: 440 nm, and rotation speed: 60 r/min; the injection volume for each catalytic luminescence test was 1 mL.

### 2.5. Gas Chromatography Method

In this study, gas chromatography (GC) was used as comparison method to verify the accuracy and practicability of the experiment. Instrument: Gas chromatograph with flame ionization detector (FID). Chromatographic column: 6% cyanopropylphenyl/94% dimethylpolysiloxane, column length: 60 m, liquid film thickness: 1.4 μm, inner diameter: 0.25 mm; carrier gas: N$_2$, column temperature: first, the temperature was kept at 50 °C for 15 min, and then increased to 150 °C at a rate of 10 °C/min, and finally held for 1 min.

### 2.6. Sample Preparation

A 1 L Teflon sample bag was used to prepare MTBE gas with different levels of purity (the concentrations were 100 mg/L). The luminescence signals from different samples were measured using the proposed CTL method. A standard addition recovery experiment was carried out by adding different amounts of MSBE, and the recovery rate was calculated.

## 3. Results

### 3.1. Synthesis and Characterization of Pt/Au NPs@Co$_3$O$_4$

#### 3.1.1. Synthesis of Pt/Au NPs@Co$_3$O$_4$

The process for manufacturing Pt/Au NPs@Co$_3$O$_4$ is shown in Figure 1. Hollow dodecahedron Co$_3$O$_4$ was synthesized using ZIF-67 as a template, then Pt/Au NPs@Co$_3$O$_4$ was synthesized by stepwise modification.

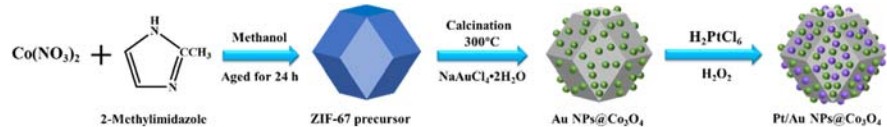

**Figure 1.** Schematic diagram of Pt/Au NPs@Co$_3$O$_4$ synthesis.

The properties and applications of Pt/Au NPs@Co$_3$O$_4$ were explored using the following techniques. SEM and TEM were adopted to characterize the morphology and structure of Pt/Au NPs@Co$_3$O$_4$. As shown in Figure 2A,B, the appearance of the Pt/Au NPs@Co$_3$O$_4$ presented as a regular polyhedral shape with an average size of about 500 nm. It is worth noting that there was a distinct hollow structure. In addition, Pt/Au NPs@Co$_3$O$_4$ had a rough surface, and it can be clearly seen that Pt NPs and Au NPs were uniformly distributed on the surface of the Co$_3$O$_4$ without changes to its morphology and structure. The TEM image showed the hollow structure and the morphology of the regular polyhedrons more clearly (Figure 2C). The existence of lattice fringes of two types of nanoparticles in the high magnification TEM image (Figure 2D) also indicated the successful recombination of Pt NPs and Au NPs. The nanoparticles were approximately 5 nm in size.

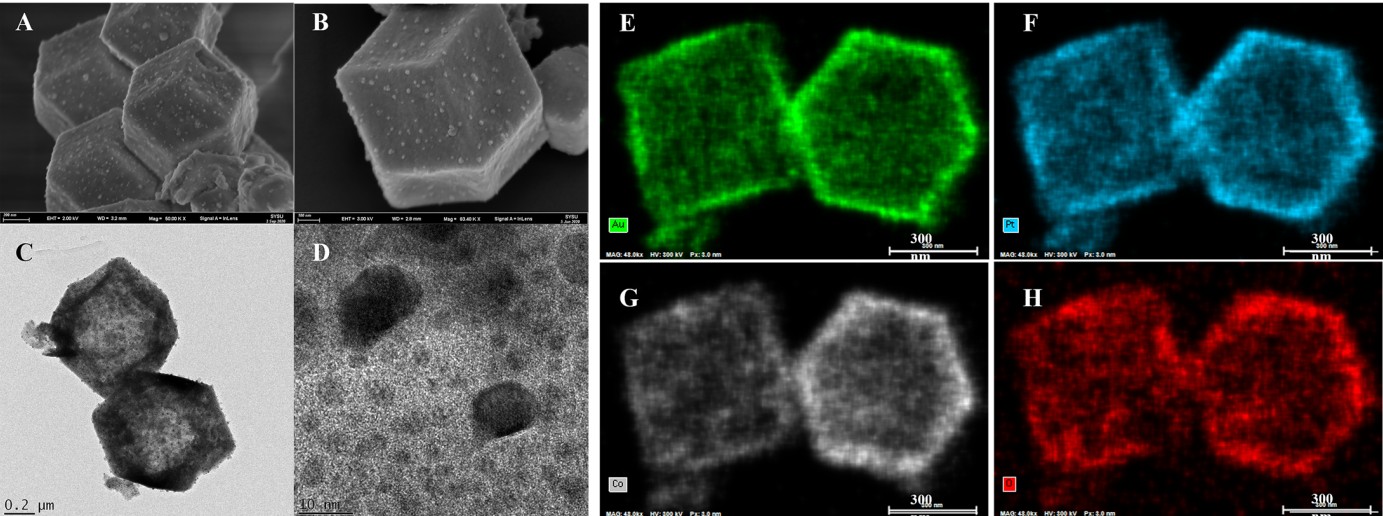

**Figure 2.** SEM (**A**,**B**) and TEM (**C**,**D**) images of Pt/Au NPs@Co$_3$O$_4$; Au (**E**), Pt (**F**), Co (**G**) and O (**H**) element mapping for Pt/Au NPs/Co$_3$O$_4$.

To characterize the distribution of elements in Pt/Au NPs@Co$_3$O$_4$ intuitively, the distribution maps of Pt, Au, Co, and O in Pt/Au NPs@Co$_3$O$_4$ were investigated using energy-dispersive X-ray spectroscopy (EDXS). It was found that Au and Pt NPs were uniformly distributed throughout the polyhedral cavity structure (Figure 2E,F). In addition, Figure 2G,H indicates the uniform distribution of Co and O in the Co$_3$O$_4$ cavity structure. The element mapping proved the successful synthesis of Pt/Au NPs@Co$_3$O$_4$ composite.

#### 3.1.2. Characterization of Pt/Au NPs@Co$_3$O$_4$

XPS was a tool which is used to analyze the elemental composition and element valence states of material surfaces. Figure 3 shows the XPS spectroscopy of Pt/Au NPs@Co$_3$O$_4$.

Figure 3A displays the spectrum of Pt 4f, there is a doublet characteristic peak at 73.20 eV and 76.05 eV which belongs to Pt $4f_{7/2}$ and Pt $4f_{5/2}$, respectively, indicating the existence of Pt(0) in Pt/Au NPs@$Co_3O_4$ [25,26]. The XPS spectrum of Au 4f also shows double peaks (Figure 3B), and the peaks at 87.90 eV and 84.20 eV belong to $4f_{5/2}$ and $4f_{7/2}$ of Au(0) [27]. In addition, the XPS report documented that the atomic% of Pt 4f and Au 4f was 2.69% and 1.25%, respectively, and the ratio of Pt to Au NPs was approximately 2:1. Figure 3C shows the XPS spectrum of Co 2p; there are two characteristic peaks that come from the spin-orbit interaction of $2p_{3/2}$ and $2p_{1/2}$ [28]. In this spectrum, the peak attributed to $2p_{3/2}$ was split into two peaks at 781.3 and 783.2 eV, while the peak for $2p_{1/2}$ was also split into two peaks with binding energies of 795.9 and 798.4 eV, respectively. The peaks at 781.3 eV (Co $2p_{3/2}$) and 795.9 eV (Co $2p_{1/2}$) confirm that Co was present as $Co_3O_4$ in the composite [29,30]. Moreover, the full-range XPS spectra of the Pt/Au NPs@$Co_3O_4$ composite confirmed the presence of all the expected elements, such as Pt, Au, C, O, and Co (Figure 3D). In summary, the results of XPS spectra proved the successful synthesis of Pt/Au NPs@$Co_3O_4$ composite.

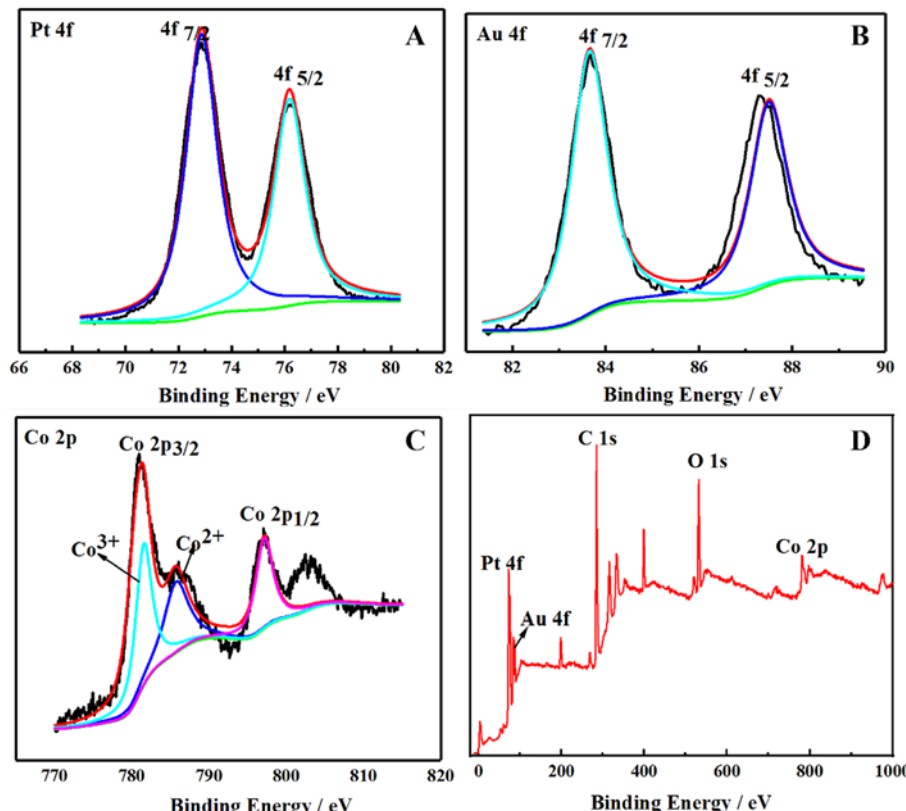

**Figure 3.** XPS spectra of Pt 4f (**A**), Au 4f (**B**), Co 2p (**C**), and Pt/Au NPs@$Co_3O_4$ (**D**).

XRD was used to characterize the crystalline structure of Pt/Au NPs@$Co_3O_4$. As shown in Figure 4A, the diffraction peaks at 2° at 19.00°, 36.85°, 44.81°, and 65.24° corresponded to the (111) face, (311) face, (400) face, and (440) face in $Co_3O_4$ respectively [31]. However, there were no diffraction peaks of Pt or Au NPs found in the XRD spectrum. The possible reason for that might be the low content and high dispersion of the Pt and Au NPs in $Co_3O_4$ [10,31].

The functional group information for Pt/Au NPs@$Co_3O_4$ was characterized using FT-IR, as shown in Figure 4B. The peaks at 3411 $cm^{-1}$ and 1610 $cm^{-1}$ are caused by the stretching and in-plane bending vibration of the N−H bond. The peaks near 2930 $cm^{-1}$, 2850 $cm^{-1}$. and 1087 $cm^{-1}$ are the stretching vibration and in-plane bending vibration peaks of the C−H bond. The vibrational peaks of the Co−O bond are associated with the two peaks located at 670 $cm^{-1}$ and 569 $cm^{-1}$ [32,33].

The specific surface area and pore size distribution of the Pt/Au NPs@Co$_3$O$_4$ composites were investigated by N$_2$ adsorption–desorption experiments at 77 K, and the results are shown in Figure 4C,D. According to the Brunauer–Emmett–Teller (BET) method, the BET specific surface area of Pt/Au NPs@Co$_3$O$_4$ was calculated to be 33.7 m$^2$/g, and the N$_2$ adsorption–desorption curve was a typical IV-type curve with an H3-type hysteresis loop, indicating the existence of a mesoporous/macroporous structure within Pt/Au NPs@Co$_3$O$_4$. Furthermore, the pore size distribution map exhibited a broad pore size, ranging from 1.6 to 30 nm, which further demonstrates the coexistence of mesopores and macropores in the material.

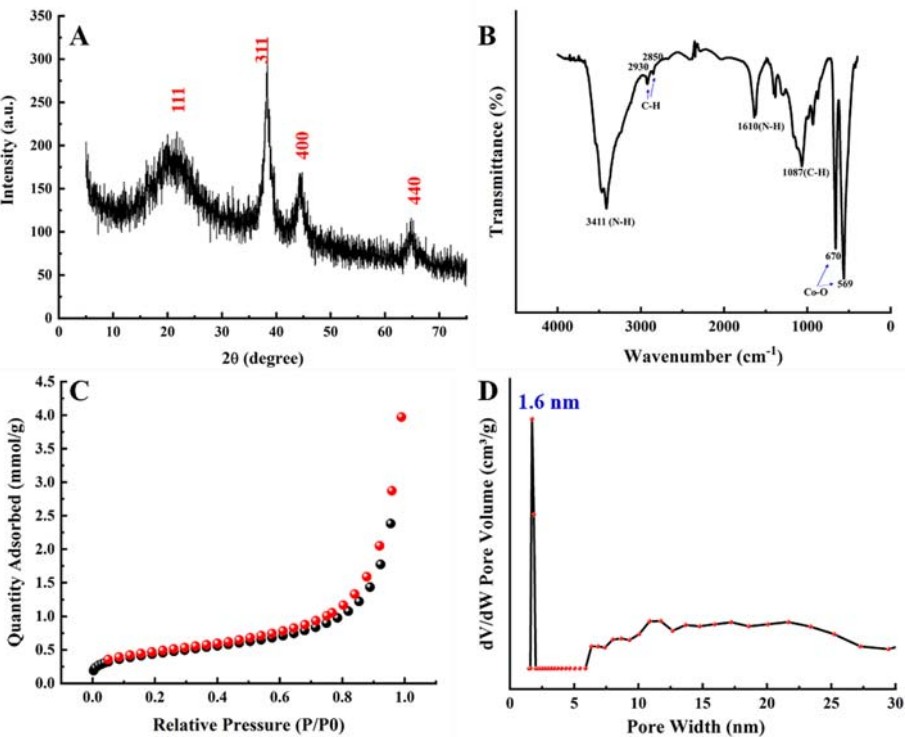

**Figure 4.** XRD patterns (**A**), FT-IR spectra (**B**), N$_2$ adsorption (black) and desorption (red) isotherms (**C**), and the corresponding pore size distribution curves (**D**) of Pt/Au NPs@Co$_3$O$_4$.

### 3.2. Cataluminescence Sensor

#### 3.2.1. CTL Signal Enhancement by Pt/Au NPs@Co$_3$O$_4$

The enhancement of the Pt/Au NPs@Co$_3$O$_4$ composite material to the MSBE CTL signal was explored by comparing it with that obtained from Co$_3$O$_4$, Au NPs@Co$_3$O$_4$, and Pt NPs@Co$_3$O$_4$. As shown in Figure 5A, the signal obtained from 5.00 mg/L MSBE was approximately 600 on the surface of Co$_3$O$_4$ (line 1), 1200 on Au NPs@Co$_3$O$_4$ (line 2), and 2000 on Pt NPs@Co$_3$O$_4$ (line 3); while the signal increased to 6500 on Pt/Au NPs@Co$_3$O$_4$ (line 4). The results suggest that both single Au NPs and single Pt NPs could improve the catalytic ability of Co$_3$O$_4$ to a certain extent. However, the catalytic performance was greatly promoted when both were present at the same time, a feature which may be caused by the electron interaction between the Au and Pt NPs [21,22].

The CTL responses of different concentrations of MSBE and MTBE on the surface of Pt/Au NPs@Co$_3$O$_4$ were also explored. As shown in Figure 5B,C, MSBE exhibited a strong CTL signal. In Figure 5B, the CTL signal was approximately 1100 when the concentration was 1.00 mg/L (curve 1), which was enhanced to approximately 5000 and 9000 when the concentration was increased to 5.00 and 10.0 mg/L (curves 2 and 3, respectively). Meanwhile, the background noise was very low, only approximately 100, resulting in a high signal-to-noise ratio. Therefore, Pt/Au NPs@Co$_3$O$_4$ showed high sensitivity for the detection of MSBE. At the same time, the CTL responses of Pt/Au NPs@Co$_3$O$_4$ to different

concentrations of MTBE were explored. As shown in Figure 5C, there was no CTL signal observed with concentrations ranging from 5.00 to 100.0 mg/L. This phenomenon shows that Pt/Au NPs@Co₃O₄ could not catalyze the CTL reaction of MTBE, which laid the foundation for establishing a CTL method based on Pt/Au NPs@Co₃O₄ that can be used to determine MSBE in MTBE and to monitor the purity of MTBE.

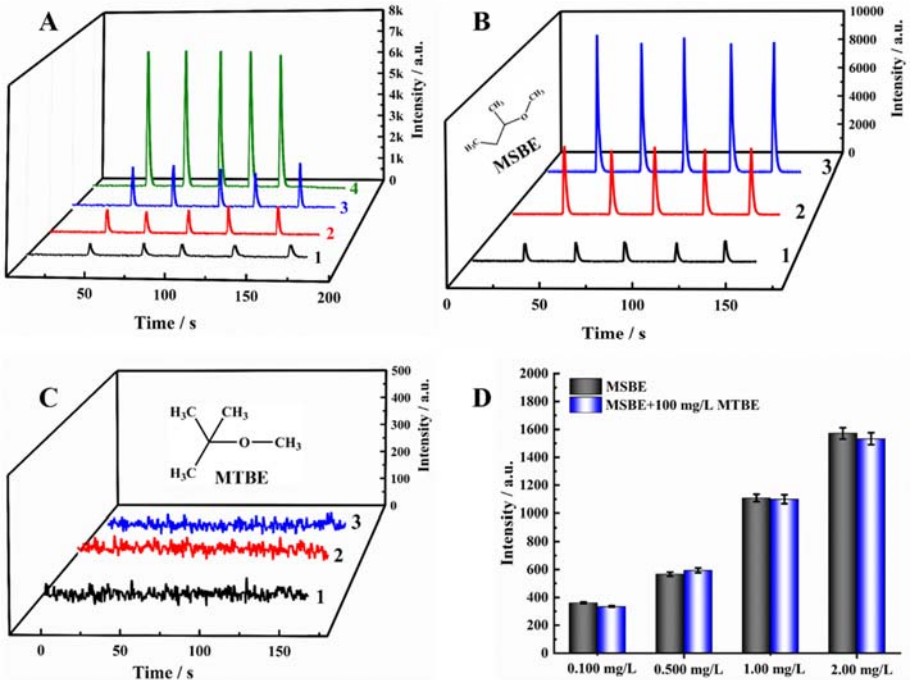

**Figure 5.** (**A**) CTL curves of 5.00 mg/L MSBE based on Co₃O₄ (line 1), Au NPs@Co₃O₄ (line 2), Pt NPs@Co₃O₄ (line 3), and Pt/Au NPs@Co₃O₄ (line 4); (**B**) CTL response of different concentrations of MSBE: 1.00 mg/L (line 1), 5.00 mg/L (line 2), and 10.0 mg/L (line 3); (**C**) CTL response of different concentrations of MTBE: 5.00 mg/L (line 1), 10.0 mg/L (line 2), and 100 mg/L (line 3); (**D**) CTL signal comparison between different concentrations of MSBE alone and with 100 mg/L MTBE addition.

### 3.2.2. The Catalytic Performance of the Cataluminescence Sensor

It is necessary that the presence of large concentrations of MTBE should not interfere with the CTL detection of MSBE if we want to determine the purity of MTBE using a CTL sensor based on Pt/Au NPs@Co₃O₄. In order to explore the anti-interference performance of Pt/Au NPs@Co₃O₄, the CTL signals of different concentrations of MSBE in the presence of 100 mg/L MTBE were detected, and compared with the detection results of MSBE alone, as shown in Figure 5D. CTL signals were measured for 0.100, 0.500, 1.00, and 2.00 mg/L MSBE in the presence of 100 mg/L MTBE (blue). Comparing these data to the results from MSBE tested alone (black), it was found that the presence of MTBE did not significantly affect the CTL signal of MSBE, including at higher concentrations. This suggested that MTBE does not interfere with the CTL response of MSBE when the content of MSBE exceeds 5%. The purity of MTBE used for gasoline octane blending must exceed 96% (wt%). Therefore, the CTL sensor based on Pt/Au NPs@Co₃O₄ could be used to determine the content of MSBE in MTBE, thus determining the purity grade of MTBE.

The possible mechanism of Pt/Au NPs@Co₃O₄ to the selective catalysis of MSBE was speculated by comparing the structures of MTBE and MSBE. The CTL test (Figure 5) showed that MSBE presented a strong CTL response, while MTBE showed no signal, therefore the functional groups involved in the CTL signaling response will not be methoxyl groups, which are present in both MTBE and MSBE [34,35]. The possible CTL reaction process of MSBE is shown as a response equation in Figure 6. The functional group that reacts is specific to MSBE, and is absent from MTBE. Thus, it is possible for the two isomers to have

completely different reactions. To support this speculation, the reaction products of MSBE with different reaction times were analyzed by gas chromatography–mass spectrometry, as shown as Figure 6. Methyl acetate (Peak 5) was found in the intermediate products of the reaction, and all the intermediate products generated by the CTL reaction were finally converted into carbon dioxide (Peak 1) after 300 s. Although the amount of methyl acetate seems very low, the emergence of methyl acetate proved the accuracy of our speculated reaction path. Methyl acetate seems to be present as an intermediate which was eventually converted to $CO_2$. However, because the sampling process was not a completely closed environment, the intensities of the peaks could not be quantified. The discovery of methyl acetate confirmed the reaction pathway shown in Figure 6, and given the CTL phenomena, it was inferred that the CTL signal of MSBE was caused by the reaction of $-CH_2-CH_3$. Thus, this explains the different CTL phenomena exhibited by MTBE and MSBE, which are isomers.

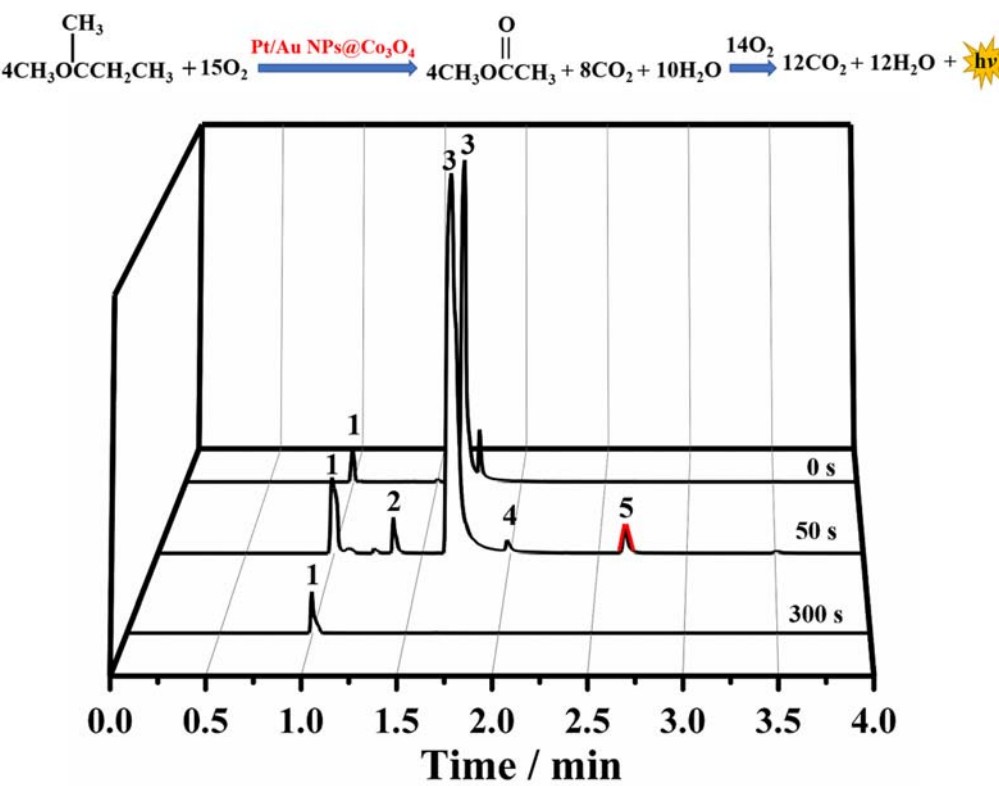

**Figure 6.** Possible mechanism of MSBE CTL oxidation reaction. Peak 1: $CO_2$, Peak 2: HOCOCH₂CH₃ , Peak 3: CH₃OCCH₂CH₃ , Peak 4: CH₃OCCH₂CH₃ , Peak 5: CH₃OCCH₃ .

### 3.3. Analytical Characteristics of the Sensor

The analytical performance evaluation, including the assessment of selectivity, stability, linear range, limit of detection, and sample analysis application, was carried out with the optimized operational conditions as follows: temperature: 187 °C, wave length: 440 nm, and rotational speed: 60 r/min.

### 3.3.1. Selectivity and Stability

The selectivity and stability of CTL sensors was the premise of establishing the analytical method. To explore the selectivity of Pt/Au NPs@$Co_3O_4$-based CTL sensors, other impurities that may coexist in MTBE were selected as interferents, including methanol, tert-butanol, propanol, acetone, and n-hexane. Then, 10.0 mg/L MSBE gas and 200 mg/L coexistence impurity gases were prepared individually in Teflon sampling bags, and the CTL signals were collected under the same operational conditions, as shown in Figure 7A. It

was found that 10 mg/L MSBE had a strong response with a signal value of approximately 9000, while the coexistence impurities showed no obvious response, indicating the excellent selective catalytic chemiluminescence capability of Pt/Au NPs@Co$_3$O$_4$ towards MSBE. In addition, the stability of the method was also explored, and the result is shown in Figure 7B. The Pt/Au NPs@Co$_3$O$_4$-based CTL gas sensor was used to detect 10.0 mg/L MSBE nine times in parallel; the results presented a low RSD value of 2.5%. In addition, the sensor was placed naturally in the air at room temperature for more than a month. The results suggested that the sensor has a good stability for a long period of use and the humidity of the air had no significant effect on the test results.

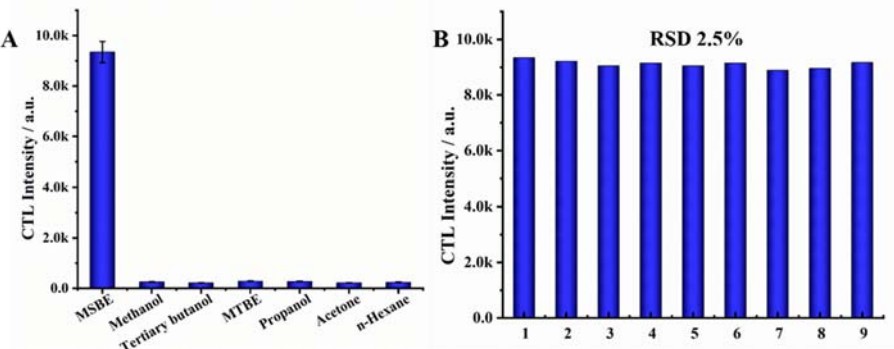

**Figure 7.** The selectivity (**A**) and repeatability (**B**) of 10 mg/L MSBE at Pt/Au NPs@Co$_3$O$_4$.

### 3.3.2. Linear Range and Limit of Detection

Under the optimized operational conditions, the CTL response curves of standard MSBE gas at concentrations of 0.100, 0.500, 1.00, 5.00, 10.0, 20.0, 30.0, 40.0, 50.0, 60.0, 70.0, 80.0, and 90.0 mg/L were recorded and are shown in Figure 8A. Figure 8B demonstrates a linear increase in the signal intensity with the MSBE concentration. Therefore, the linear range of the Pt/Au NPs@Co$_3$O$_4$-based CTL analysis method was 0.100–90.0 mg/L, and the limit of detection was calculated as 0.031 mg/L ($S/N = 3$).

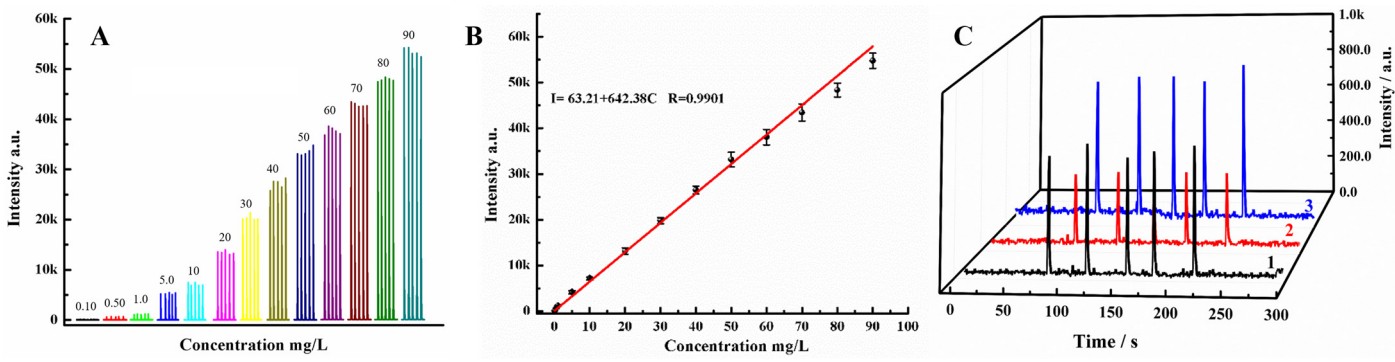

**Figure 8.** (**A**) CTL response of different concentrations, (**B**) the calibration plots of MSBE versus signal intensity (insert), and (**C**) CTL curves of 0.50 mg/L MSBE (line 1), 95% (wt%) MTBE samples (line 2), and after the addition of 0.50 mg/L (line 3).

### 3.3.3. Sample Analysis

To evaluate its practicability, the Pt/Au NPs@Co$_3$O$_4$-based CTL analysis method was employed to detect the content of MSBE in MTBE gasoline additives with different purity grades, including 95%, 98%, 99%, and >99.9% (wt%). Thus, 100 mg/L MTBE gas samples were prepared in Teflon sampling bags, and the CTL curve for each sample was recorded. The results showed 0.31 mg/L MSBE was found in the 95% (wt%) pure MTBE gas sample, and its CTL curve (line 2) is shown in Figure 8C; while no CTL signal was observed in the other samples. These results indicate that 95% (wt%) MTBE is not suitable for gasoline octane blending, while the content of MSBE in 98%, 99%, and >99.9% (wt%) MTBE was

less than 0.1% (wt%). Moreover, the results of the recovery tests which were carried out by adding 0.50, 1.00, and 3.00 mg/L MSBE were in the range of 92.0–108.6%. In addition, these four MTBE gas samples were also analyzed using GC. A relative error of 3.7% was obtained between the results of the proposed CTL method and the standard GC method, indicating the accuracy and reliability of the Pt/Au NPs@$Co_3O_4$-based CTL analysis method. The detailed results of MTBE gas samples analysis are listed in Table 1.

**Table 1.** Detection and recovery of MSBE in MTBE samples.

| Sample (wt%) | The CTL Method | | | | | | GC (mg/L) | Relative Error (%) |
|---|---|---|---|---|---|---|---|---|
| | Amount (mg/L) | RSD (%) ($n$ = 5) | Recovery (%) | | | | | |
| | | | 0.500 (mg/L) | 1.00 (mg/L) | 3.00 (mg/L) | | | |
| 95% MTBE | 0.31 | 2.6 | 106.2 | 92.0 | 94.4 | | 0.29 ± 0.013 | +3.7 |
| 98% MTBE | ND | - | 92.0 | 98.0 | 108.6 | | ND | - |
| 99% MTBE | ND | - | 106.8 | 93.0 | 99.4 | | ND | - |
| 99.9% MTBE | ND | - | 103.5 | 93.4 | 105.3 | | ND | - |

ND: Not detected.

## 4. Conclusions

MSBE is the main by-product of the industrial production of MTBE, and affects its application in gasoline octane blending. In this work, a Pt/Au NPs@$Co_3O_4$ CTL sensor was developed for rapid analysis of MSBE as an impurity in MTBE gasoline additives. A hollow dodecahedral structure $Co_3O_4$ material was synthesized successfully using MOFs as a template, while the Pt/Au NPs@$Co_3O_4$ composite was synthesized via a two-step method. The synthesized Pt/Au NPs@$Co_3O_4$ material exhibited excellent selective catalytic chemiluminescence properties towards MSBE, even in the presence of a high concentration of MTBE. Thus, a Pt/Au NPs@$Co_3O_4$ CTL sensor was constructed for rapid analysis of MSBE. The CTL method for MSBE analysis had a linearity range of 0.10–90 mg/L and a LOD of 0.031 mg/L ($S/N$ = 3). Finally, this Pt/Au NPs@$Co_3O_4$ CTL sensor was applied to MSBE analysis in MTBE gasoline additives with different purity grades. The results were in good agreement with those obtained from gas chromatography, and satisfactory recoveries were achieved, in the range of 92.0–109.2%. Overall, the developed Pt/Au NPs@$Co_3O_4$-based CTL method was accurate and reliable, and can be used for rapid analysis of the purity of MTBE gasoline additives, such as for quality monitoring.

**Author Contributions:** Z.S.: Conceptualization, Methodology, Data Curation, Investigation, Writing-original draft preparation. L.X.: Visualization, Writing—review & editing. G.L.: Visualization, Resources, Funding Acquisition, Supervision, Project Administration, Writing—review & editing. Y.H.: Visualization, Funding Acquisition. All authors have read and agreed to the published version of the manuscript.

**Funding:** This research was funded by the State Key Program of National Natural Science of China (No. 22134007), the National Natural Science Foundation of China (No. 21976213), the National Key Research and Development Program of China (No. 2019YFC1606101), the Research and Development Plan for Key Areas of Food Safety in Guangdong Province of China (No. 2019B020211001), and the Opening Project of National and Local Joint Engineering Research Center for Mineral Salt Deep Utilization, China (No. SF202001), respectively.

**Institutional Review Board Statement:** Not applicable.

**Informed Consent Statement:** Not applicable.

**Data Availability Statement:** Not applicable.

**Conflicts of Interest:** The authors declare no conflict of interest.

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
