# Peer review of "Pt/Au Nanoparticles@Co3O4 Cataluminescence Sensor for Rapid Analysis of Methyl Sec-Butyl Ether Impurity in Methyl Tert-Butyl Ether Gasoline Additive"

_chemosensors, doi:10.3390/chemosensors10070260_

Round 1
Reviewer 1 Report
In “ Pt/Au nanoparticles@Co3O4 cataluminescence sensor for rapid analysis of methyl sec-butyl ether impurity in methyl tert-butyl ether gasoline additive”, the authors explore the cataluminescence properties of Co3O4 nanostructures decorated by Au/Pt nanoparticles. In particular, they focused on the possibility to detect and quantify the amount of MSBE in MTBE since this amount should not exceed 5%. Linear range and selectivity have been evaluated and compared to pristine materials. Finally, the authors proposed a possible mechanism for the catalysis of MSBE according to the results obtained by GC-MS.
The work is in general well-conducted even if some points should be elucidated or described with more rigor. Here below a list of points that should be amended or commented:
1 Authors utilized an ultraweak luminescent analyzer to conduct their experiments. It would be helpful if they can comment on the possibility to detect the cataluminescence signals also with portable platforms or a low-cost setup (usually sensors require the possibility to be portable and low cost to operate in situ.)
2 There are many experimental details missing in the experimental paragraph, such as the working temperature, the emission wavelength, and an accurate description of how vapor samples were obtained. Please provide all the necessary information to accurately repeat the experiments.
3 Albeit the authors showed good repeatability of responses, the reviewer was wondering about the reproducibility. In the manuscript, often the results differ from each other greatly. For example, considering the response to 1.00 mg/L of MSBE, In figure 5A curve 1 is about 2600, whereas in Figure 5D the response is around 800 ( with and without MTBE). For example, the response to 10 mg/L in Figure 7A is around 9000, in the figure, 7B is around 7500, and in figure 5 is around 10000. Please clarify if these differences are due to a different setup or to the reproducibility in the fabrication procedure. In this latter case please comment on possible remediations.
4 Considering the GC measurements, all the peaks should be assigned. Moreover, it should be discussed if they are relevant to the proposed catalytic mechanism.
5 Furthermore, CO2 is found also in the sample at time = 0s. It is from ambient air? From the 3D graph is not easy to have the evaluation of intensities of peaks if some of them are already in the background. From the graphs, the amount of methyl acetate seems very low and it seems to be always present with reagents ( MSBE). The proposed catalytic path seems to be more likely an alternative path rather than the main catalytic reaction mechanism, considering also the high amount of MSBE. Please comment more accurately the GC results obrained.
6 The reaction proposed in Figure 6 is not balanced. Please provide a balanced reaction for each step.
7 Did the authors test the selectivity to other compounds ( see Fig.7) in the case of PtNPs@Co3O4 and AuNPs@Co3O4 ?
8 is it possible to estimate the coating percentage of Au and Pt? For example by elementary analysis considering the atomic or molecular percentage of these metals?
9 A state of the art of sensors for the detection of MSBE should be provided.
10 Scale bars in Figures A) and B) are not clearly visible
11 It should be better to describe the resistance of ultrapure water in MΩ·cm
Author Response
Response to reviewer 1
Comments: In “Pt/Au nanoparticles@Co3O4 cataluminescence sensor for rapid analysis of methyl sec-butyl ether impurity in methyl tertbutyl ether gasoline additive”, the authors explore the cataluminescence properties of Co3O4 nanostructures decorated by Au/Pt nanoparticles. In particular, they focused on the possibility to detect and quantify the amount of MSBE in MTBE since this amount should not exceed 5%. Linear range and selectivity have been evaluated and compared to pristine materials. Finally, the authors proposed a possible mechanism for the catalysis of MSBE according to the results obtained by GC-MS. The work is in general well-conducted even if some points should be elucidated or described with more rigor. Here below a list of points that should be amended or commented:
Response: We very much appreciate the careful reading of our manuscript and the valuable comments. The comments made by the referee were explained point-to-point and some important issues were addressed as following.
Specific comments:
Comment 1: Authors utilized an ultraweak luminescent analyzer to conduct their experiments. It would be helpful if they can comment on the possibility to detect the cataluminescence signals also with portable platforms or a low-cost setup (usually sensors require the possibility to be portable and low cost to operate in situ.)
Response: Thank you very much for your comment. The portable platforms or a low-cost setup are indeed more helpful to operate in situ. In our follow-up research work, we will focus more on the development of portable platforms or a low-cost setup. Thank you again for your valuable suggestions.
Comment 2: There are many experimental details missing in the experimental paragraph, such as the working temperature, the emission wavelength, and an accurate description of how vapor samples were obtained. Please provide all the necessary information to accurately repeat the experiments.
Response: Thank you very much for the suggestion. The experimental details including detecting conditions and how vapor samples were obtained were added in Marked manuscript, shown as lines 130-133. Thank you for the suggestion.
Comment 3: Albeit the authors showed good repeatability of responses, the reviewer was wondering about the reproducibility. In the manuscript, often the results differ from each other greatly. For example, considering the response to 1.00 mg/L of MSBE. In figure 5A curve 1 is about 2600, whereas in Figure 5D the response is around 800 (with and without MTBE). For example, the response to 10 mg/L in Figure 7A is around 9000, in the figure, 7B is around 7500, and in figure 5 is around 10000. Please clarify if these differences are due to a different setup or to the reproducibility in the fabrication procedure. In this latter case please comment on possible remediations.
Response: We very much appreciate the careful reading of our manuscript and the valuable comments. Curve 1 in figure 5A corresponded to the CTL signal of 5.00 mg/L MSBE on Co3O4 surface, while Figure 5D was the CTL signal of 1.00 mg/L MSBE on Pt/Au NPs@Co3O4 surface. Therefor, the CTL signals were different. The reason for the different of 10 mg/L MSBE signal in Figure 7A, and Figure 5 was whether the background noise is removed or not. Besides, the data in Figure 7B was obtained before conditional optimization. It is our responsibility and the data is not handled in a uniform manner. The data in the Figure 7B was retested under the optimization conditions, shown as Figure 7B in Marked manuscript.
Comment 4: Considering the GC measurements, all the peaks should be assigned. Moreover, it should be discussed if they are relevant to the proposed catalytic mechanism.
Response: Thank you very much for your suggestion. All the peaks have been assigned in Fig.6. Because of the other peaks were not relevant to the proposed catalytic mechanism, we did not discuss them in manuscript. Thank you again for the suggestion.
Comment 5: Furthermore, CO2 is found also in the sample at time =0s. It is from ambient air? From the 3D graph is not easy to have the evaluation of intensities of peaks if some of them are already in the background. From the graphs, the amount of methyl acetate seems very low and it seems to be always present with reagents (MSBE). The proposed catalytic path seems to be more likely an alternative path rather than the main catalytic reaction mechanism, considering also the high amount of MSBE. Please comment more accurately the GC results obtained.
Response: We very much appreciate the careful reading of our manuscript and the valuable comments. Because the test process was carried out under the condition of air as carrier gas, the CO2 in the sample at time =0 s was from air. Besides, the reason for the amount of methyl acetate seems very low was that it's an intermediate, which was eventually converted to CO2. The emergence of methyl acetate proved the correctness of our speculated reaction path. But because the sampling process was not a completely closed environment, the intensities of peaks could not be quantified. The GC results obtained were commented more accurately in Marked manuscript, shown as line 270-274. In the follow-up research work, we will be very willing to improve the design of the device and optimize the whole process to investigate the reaction mechanism in more detail, systematically. Thank you again for the suggestion.
Comment 6: The reaction proposed in Figure 6 is not balanced. Please provide a balanced reaction for each step.
Response: Thank you for the suggestion. According to the opinion, we have balanced reaction for each step in Fig.6. Shown as Marked manuscript. Thank you again for the suggestion.
Comment 7: Did the authors test the selectivity to other compounds (see Fig.7) in the case of PtNPs@Co3O4 and AuNPs@Co3O4?
Response: We very much appreciate the careful reading of our manuscript and the comments. In this paper, we established the CTL analytical method for MSBE detection based on Pt/Au NPs@Co3O4, and tested the selectivity of Pt/Au NPs@Co3O4 to other compounds (shown as Fig.7A). Because the CTL sensor was constructed based on Pt/Au NPs@Co3O4, we didn’t test the selectivity to other compounds in the case of PtNPs@Co3O4 and AuNPs@Co3O4. Thank you very much!
Comment 8: Is it possible to estimate the coating percentage of Au and Pt? For example, by elementary analysis considering the atomic or molecular percentage of these metals?
Response: Thank you for the suggestion. It is possible to estimate the coating percentage of Au and Pt from XPS data. In XPS report, Atomic% of Pt 4f and Au 4f were 2.69% and 1.25%, the ratio Pt NPs and Au NPs was about 2:1. The related content was added in Marked manuscript, shown as lines 182-183.
Comment 9: A state of the art of sensors for the detection of MSBE should be provided.
Response: We very much appreciate the careful reading of our manuscript and the valuable comments. The state of the art of sensors for the detection of MSBE has been added in Marked manuscript, shown as lines 40-41.
Comment 10: Scale bars in Figures 2A and Figures 2B are not clearly visible.
Response: We very much appreciate the careful reading of our manuscript. Scale bars in Figures 2A and Figures 2B have been adjusted, as shown as Fig.2 in Marked manuscript.
Comment 11: It should be better to describe the resistance of ultrapure water in MΩcm.
Response: We very much appreciate the careful reading of our manuscript. The resistance of ultrapure water was described in MΩcm, as show as line 90 in Marked manuscript.
Reviewer 2 Report
Methyl tert-butyl ether (MTBE) is an effective additive to tune gasoline octane value. However, methyl sec-butyl ether (MSBE), which is a main by-product in the industrial production of MTBE, affects the purity of MTBE. A simple and rapid method for monitoring MSBE in MTBE gasoline additive is essential for petrochemical Industry. Now Shi et al. synthesized Pt/Au NPs@Co3O4 with a hollow dodecahedron three-dimensional structure using ZIF-67 as template. A cataluminescence (CTL) sensor was established for the determination of MSBE based on the specificity of Pt/Au NPs@Co3O4. The experimental results showed that Pt/Au NPs@Co3O4 showed a strong specific CTL response to MSBE, with no interference of MTBE. The sensor was used to detect the MSBE in MTBE with different purity grades, with recoveries ranged from 92.0% to 109.2%, and analysis results were consistent with those determined by the gas chromatography. The results are of wide interest, of good significance and of fine impact. Nevertheless, there are some issues required to be clarified and thus a careful revision is needed before the manuscript could be accepted.
The following points are important to improve the quality of the manuscript and thus shuld be addressed and included in the revised manuscript.
(1) Before introduction of Co3O4 nanomaterials, it is of great significance to introduce that metal oxides are a common type of materials used for various sensors. For instances, WO3 [see: Li, J.-H.; Wu, J.; Yu, Y.-X. DFT exploration of sensor performances of two-dimensional WO3 to ten small gases in terms of work function and band gap changes and I-V responses. Appl. Surf. Sci. 2021, 546, 149104.], ZnO [see: Singh, P.; Simanjuntak, F.M.; Hu, L.-L.; Tseng, T.-Y.; Zan, H.-W.; Chu, J.P. Negative effects of annealed seed layer on the performance of ZnO-nanorods based nitric oxide gas sensor. Sensors 2022, 22, 390.], SnO2 [see: Goto, T.; Itoh, T.; Akamatsu, T.; Izu, N.; Shin, W. CO sensing properties of Au/SnO2-Co3O4 catalysts on a micro thermoelectric gas sensor. Sensors and Actuators B-Chemical 2016, 223(1), 774-783.] and Co3O4 [see: Yoon, J.; Kim, H.; Jeong, H.; Lee, J. Gas sensing characteristics of p-type Cr2O3 and Co3O4 nano-fibers depending on inter-particle connectivity. Sensors and Actuators B-Chemical 2014, 202(1): 263-271.] have been frequently used as sensor materials. Introduction of above typical metal oxide sensor materials is helpful for readers to understand why authors select Co3O4 as a support of noble metals in the sensor and thus improve the logic and significance of the manuscript.
(2) For comparison the XRD pattern of pure CO3O4 should also be included in Figure 4A to demonstrate that Pt/Au nanoparticle has attached on the surface of Co3O4 nanocrystal. In addition N2 adsorption and desorption isotherms should be specified by a temperature. 77 K maybe the case in the manuscript.
(3) The stability and durability of the developed cataluminescence sensor based on Pt/Au NPs@Co3O4 should also be discussed and predicted for a long period use (for example, how many months or years) as well as the effect of humidity of air in the main text of the manuscript.
(4) It seems that the chemical formulae of MSBE is incorrect in Figure 6. Please check it.
(5) There are some grammar errors and/or typos. For instances, Line 14, “Pt NPs and Au NPs …” should be revised to “where Pt nanoparticles (NPs) and Au NPs …”. It's better to reverse the order of the two sentences in Lines 14-18. Line 40, “cataluminescence (CTL) owned many benefits” should be corrected to “cataluminescence (CTL) owns many benefits”. Lines 41-42, “Catalyst is …, which have an important effect” should be corrected to “Catalyst is …, which has an important effect”. Line 56, “noble metal () to metal oxides” should be corrected to “noble metal () to metal oxides”. Lines 57-59, “… not only prevents…, but also improve …” should be corrected to “… not only prevents…, but also improves …”. Lines 129-130, “Gas chromatography” should be corrected to “gas chromatography”. Line 262, “The found of methyl acetate” should be corrected to “The finding of methyl acetate”. “which two are isomers” should be corrected to “which are isomers”. And so forth.
(6) The format of authors in the reference List does not match what the journal requires. For instances, the first three references should be listed as
1. Chow, J. C.; Yu, J.Z.; Watson, J. G.; Ho, S. S. H.; Bohannan, T. L.; Hays, M. D.; Fung, K. K. The application of thermal methods for determining chemical composition of carbonaceous aerosols: A review. Journal of Environmental Science and Health Part a-Toxic/Hazardous Substances & Environmental Engineering 2007, 42(11), 1521-1541.
2. Zhong, Y.; Huang, W.T.; Zhang, C.; Zhang, R.K.; Hu Y.F.; Xiao, X.H.; Li, G.K. Study of cyclic cataluminescence virtual sensor array for gasoline quality monitoring. Sensors and Actuators: B. Chemical 2022, 364, 131901.
3. Seddon, D. Reformulated gasoline, opportunities for new catalyst technology. Catalysis Today 1992, 15(1), 1-21.
Author Response
Response to reviewer 2
Comments: Methyl tert-butyl ether (MTBE) is an effective additive to tune gasoline octane value. However, methyl sec-butyl ether (MSBE), which is a main by-product in the industrial production of MTBE, affects the purity of MTBE. A simple and rapid method for monitoring MSBE in MTBE gasoline additive is essential for petrochemical Industry. Now Shi et al. synthesized Pt/AuNPs@Co3O4 with a hollow dodecahedron three-dimensional structure using ZIF-67 as template. A cataluminescence (CTL)sensor was established for the determination of MSBE based on the specificity of Pt/Au NPs@Co3O4. The experimental results showed that Pt/Au NPs@Co3O4 showed a strong specific CTL response to MSBE, with no interference of MTBE. The sensor was used to detect the MSBE in MTBE with different purity grades, with recoveries ranged from 92.0% to 109.2%, and analysis results were consistent with those determined by the gas chromatography. The results are of wide interest, of good significance and of fine impact. Nevertheless, there are some issues required to be clarified and thus a careful revision is needed before the manuscript could be accepted. The following points are important to improve the quality of the manuscript and thus should be addressed and included in the revised manuscript.
Response: We appreciate very much for your nice comments and recommendation.
Specific comments:
Comment 1: Before introduction of Co3O4 nanomaterials, it is of great significance to introduce that metal oxides are a common type of materials used for various sensors. For instances, WO3 [see: Li, J.-H.; Wu, J.; Yu, Y.-X. DFT exploration of sensor performances of two-dimensional WO3 to ten small gases in terms of work function and band gap changes and I-V responses. Appl. Surf. Sci. 2021, 546, 149104.], ZnO[see: Singh, P.; Simanjuntak, F.M.; Hu, L.-L.; Tseng, T.-Y.; Zan, H.-W.; Chu, J.P. Negative effects of annealed seed layer on the performance of ZnO-nanorods based nitricoxide gas sensor. Sensors 2022, 22, 390.], SnO2 [see: Goto, T.; Itoh, T.; Akamatsu, T.; Izu, N.; Shin, W. CO sensing properties of Au/SnO2-Co3O4 catalysts on a micro thermoelectric gas sensor. Sensors and Actuators B-Chemical 2016, 223(1), 774-783.] and Co3O4 [see: Yoon, J.;Kim, H.; Jeong, H.; Lee, J. Gas sensing characteristics of p-type Cr2O3 and Co3O4 nano-fibers depending on inter-particle connectivity. Sensors and Actuators B-Chemical2014, 202(1): 263-271.] have been frequently used as sensor materials. Introduction of above typical metal oxide sensor materials is helpful for readers to understand why authors select Co3O4 as a support of noble metals in the sensor and thus improve the logic and significance of the manuscript.
Response: Thank you for your suggestion. As a common type of materials used for various sensors, metal oxides have been introduced before introduction of Co3O4 nanomaterials. As shown as lines 46-48 and Ref. [6-9] in marked manuscript.
Comment 2: For comparison the XRD pattern of pure Co3O4 should also be included in Figure 4A to demonstrate that Pt/Au nanoparticle has attached on the surface of Co3O4 nanocrystal. In addition, N2 adsorption and desorption isotherms should be specified by a temperature. 77 K maybe the case in the manuscript.
Response: Thank you for your suggestion. The XRD pattern of pure Co3O4 was same with Fig. 4A, because of the low content and high dispersion of Pt NPs and Au NPs in Co3O4, there were no diffraction peaks of Pt NPs or Au NPs were found in the XRD spectrum. Therefore, we did not include the XRD pattern of pure Co3O4 in the article. The Related explanations were shown in lines 196-198 and Ref.10 and Ref.31. The attachment of Pt/Au nanoparticle on the surface of Co3O4 nanocrystal could be demonstrated by Fig.2 and Fig.3. In XPS report, Atomic% of Pt 4f and Au 4f were 2.69% and 1.25%, the ratio Pt NPs and Au NPs was about 2:1 (shown as lines 182-183). In addition, the temperature for N2 adsorption and desorption isotherms have been specified in marked manuscript.
Comment 3: The stability and durability of the developed cataluminescence sensor based on Pt/Au NPs@Co3O4 should also be discussed and predicted for a long period use (for example, how many months or years) as well as the effect of humidity of air in the main text of the manuscript.
Response: We very much appreciate the careful reading of our manuscript and the valuable comments. The stability and durability of the developed cataluminescence sensor for a long period use and the effect of humidity of air have been discussed and predicted in the main text of the manuscript. As shown as lines 297-300 in marked manuscript.
Comment 4: It seems that the chemical formulae of MSBE is incorrect in Figure 6. Please check it.
Response: We very much appreciate the careful reading of our manuscript. The chemical formulae of MSBE in Fig. 6 has been checked and corrected.
Comment 5: There are some grammar errors and/or typos. For instances, Line 14, “Pt NPs and Au NPs …” should be revised to “where Pt nanoparticles (NPs) and Au NPs …”. It's better to reverse the order of the two sentences in Lines 14-18. Line 40, “cataluminescence (CTL) own edmany benefits” should be corrected to “cataluminescence(CTL) owns many benefits”. Lines 41-42, “Catalyst is…, which have an important effect” should be corrected to “Catalyst is …, which has an important effect”. Line 56,“noble metal () to metal oxides” should be corrected to “noble metal () to metal oxides”. Lines 57-59, “… notonly prevents…, but also improve …” should be corrected to“… not only prevents…, but also improves …”. Lines129-130, “Gas chromatography” should be corrected to “gas chromatography”. Line 262, “The found of methylacetate” should be corrected to “The finding of methylacetate”. “which two are isomers” should be corrected to “which are isomers”. And so forth.
Response: We very much appreciate the careful reading of our manuscript and the valuable comments. We also have carefully checked the manuscript for the language problems and corrected them. All of the changes were marked in Marked Manuscript.
Comment 6: The format of authors in the reference List does not match what the journal requires. For instances, the first three references should be listed as
- Chow, J. C.; Yu, J.Z.; Watson, J. G.; Ho, S. S. H.; Bohannan, T. L.; Hays, M. D.; Fung, K. K. The application of thermal methods for determining chemical composition of carbonaceous aerosols: A review Journal of Environmental Science and Health Part a-Toxic/Hazardous Substances & Environmental Engineering 2007, 42(11), 1521-1541.
- Zhong, Y.; Huang, W.T.; Zhang, C.; Zhang, R.K.; HuY.F.; Xiao, X.H.; Li, G.K. Study of cyclic cataluminescence virtual sensor array for gasoline quality monitoring. Sensors and Actuators: B. Chemical2022, 364, 131901.
- Seddon, D. Reformulated gasoline, opportunities for new catalyst technology. Catalysis Today 1992, 15(1), 1-21.
Response: We very much appreciate the careful reading of our manuscript. The format of authors in the reference List have been corrected and match what the journal requires. Thank you very much!
Round 2
Reviewer 1 Report
The reviewer thanks the authors for their responses and for having amended the manuscript accordingly with the proposed remarks. In the reviewer’s opinion, some minor issues should be further checked.
1) First of all, the reviewer wrongly reported the plots non in accordance with each other in the previous report. I would try to report here carefully the lines in the manuscript and graphs that seem to be not in accordance with each other.
On lines 226 – 229 the authors wrote:” The CTL responses of different concentrations of MSBE and MTBE on the surface of Pt/Au NPs@Co3O4 were also explored. As shown in Fig.5B and Fig.5C, MSBE showed strong CTL signal. In Fig.5B, the CTL signal was about 2300 (curve 1) when the concentration was 1.00 mg/L,”here the response of Pt/Au NPs@Co3O4 to 1 mg/L of MSBE is around 2300.
On the other hand, on lines 248-253 it is written: “In order to explore the anti-interference performance of Pt/Au NPs@Co3O4, the CTL signals of different concentrations of MSBE in the coexistence of 100 mg/L MTBE were detected, and compared with the detection results of MSBE alone, as shown in Fig.5D. The CTL signal of 0.100, 0.500, 1.00, and 2.00 mg/L MSBE with the attendance of 100 mg/L MTBE were measured (blue). In contrast to the results when MSBE was tested alone (black),” Here the response to 1 mg/L of Pt/Au NPs@Co3O4 for MSBE alone (black columns) is around 800. Please specify if different data treatments are applied.
2)Calibration curve in the inset of Figure 8A is not visible since too small. Maybe it should be reported in a dedicated panel in Figure 8.
Author Response
Response to reviewer 1
Comments: The reviewer thanks the authors for their responses and for having amended the manuscript accordingly with the proposed remarks. In the reviewer’s opinion, some minor issues should be further checked.
Response: We very much appreciate the careful reading of our manuscript and the valuable comments. The comments made by the referee were explained point-to-point and some important issues were addressed as following.
Specific comments:
Comment 1: First of all, the reviewer wrongly reported the plots non in accordance with each other in the previous report. I would try to report here carefully the lines in the manuscript and graphs that seem to be not in accordance with each other. On lines 226-229 the authors wrote:” The CTL responses of different concentrations of MSBE and MTBE on the surface of Pt/Au NPs@Co3O4 were also explored. As shown in Fig.5B and Fig.5C, MSBE showed strong CTL signal. In Fig.5B, the CTL signal was about 2300 (curve 1) when the concentration was 1.00 mg/L,” here the response of Pt/Au NPs@Co3O4 to 1 mg/L of MSBE is around 2300. On the other hand, on lines 248-253 it is written: “In order to explore the anti-interference performance of Pt/Au NPs@Co3O4, the CTL signals of different concentrations of MSBE in the coexistence of 100 mg/L MTBE were detected, and compared with the detection results of MSBE alone, as shown in Fig.5D. The CTL signal of 0.100, 0.500, 1.00, and 2.00 mg/L MSBE with the attendance of 100 mg/L MTBE were measured (blue). In contrast to the results when MSBE was tested alone (black),” Here the response to 1 mg/L of Pt/Au NPs@Co3O4 for MSBE alone (black columns) is around 800. Please specify if different data treatments are applied.
Response: We very much appreciate the careful reading of our manuscript and the valuable comments. I'm sorry to make such careless mistakes in the manuscript. The data in Fig 5D has been reprocessed by adding background noise. Besides, the raw data of Fig.5B have been examined, it was found that the CTL signal about 2300 (curve 1) was not correspond to 1.00 mg/L, and the data has been replaced by the CTL signal of 1.00 mg/L. Thank you again for the suggestion.
Comment 2: Calibration curve in the inset of Figure 8A is not visible since too small. Maybe it should be reported in a dedicated panel in Figure 8.
Response: Thank you very much for the suggestion. The inset of Fig. 8A has been be reported in a dedicated panel, shown as Fig. 8B in marked manuscript.
Reviewer 2 Report
The authors have revised their manuscript carefully by fully considering my comments and suggestions. Now I believe the manuscript has been sufficiently improved to warrant publication in Chemosensors. I strongly recommend the revised manuscript for publication.
Author Response
Response to reviewer 2
Comments: The authors have revised their manuscript carefully by fully considering my comments and suggestions. Now I believe the manuscript has been sufficiently improved to warrant publication in Chemosensors. I strongly recommend the revised manuscript for publication.
Response: We appreciate very much for your nice comments and recommendation.
Round 3
Reviewer 1 Report
The authors exhaustively commented on all the reviewer's remarks and in the reviewer's opinion, the work is worth being published in its current form.